# The unfolded protein response in fission yeast modulates stability of select mRNAs to maintain protein homeostasis

Philipp Kimmig[1‡], Marcy Diaz[1‡], Jiashun Zheng[1], Christopher C Williams[1†a], Alexander Lang[1†b], Tomas Aragón[1†c], Hao Li[1], Peter Walter[2*]

[1]Department of Biochemistry and Biophysics, University of California, San Francisco, San Francisco, United States; [2]Department of Biochemistry and Biophysics, Howard Hughes Medical Institute, University of California, San Francisco, San Francisco, United States

**\*For correspondence:** peter@walterlab.ucsf.edu

**†Present address:** [a]Department of Cellular and Molecular Pharmacology, University of California, San Francisco, United States; [b]ETH Zürich, Institute of Biochemistry, Zürich, Switzerland; [c]Department of Hepatology and Gene Therapy, Center of Applied Medical Research, Pamplona, Spain

**‡**These authors contributed equally to this work

**Competing interests:** The authors have declared that no competing interests exist

**Reviewing editor**: Roy Parker, University of Colorado at Boulder, United States

**Abstract** The unfolded protein response (UPR) monitors the protein folding capacity of the endoplasmic reticulum (ER). In all organisms analyzed to date, the UPR drives transcriptional programs that allow cells to cope with ER stress. The non-conventional splicing of *Hac1* (yeasts) and *XBP1* (metazoans) mRNA, encoding orthologous UPR transcription activators, is conserved and dependent on Ire1, an ER membrane-resident kinase/endoribonuclease. We found that the fission yeast *Schizosaccharomyces pombe* lacks both a Hac1/XBP1 ortholog and a UPR-dependent-transcriptional-program. Instead, Ire1 initiates the selective decay of a subset of ER-localized-mRNAs that is required to survive ER stress. We identified *Bip1* mRNA, encoding a major ER-chaperone, as the sole mRNA cleaved upon Ire1 activation that escapes decay. Instead, truncation of its 3′ UTR, including loss of its polyA tail, stabilized *Bip1* mRNA, resulting in increased Bip1 translation. Thus, *S. pombe* uses a universally conserved stress-sensing machinery in novel ways to maintain homeostasis in the ER.

## Introduction

Homeostatic control mechanisms are essential to life, allowing cells to balance capacity and demand of numerous physiological processes. One such mechanism, the unfolded protein response (UPR), operates in all eukaryotic cells to adjust the protein folding capacity of the endoplasmic reticulum (ER) according to need. Environmental or physiological demands can lead to an imbalance between the protein folding load and the protein folding capacity in the ER lumen, resulting in an accumulation of unfolded or misfolded proteins, a condition termed 'ER stress' (*Walter and Ron, 2011*). When unmitigated, ER stress is toxic to cells and triggers cell death (*Shore et al., 2011*; *Tabas and Ron, 2011*; *Hetz, 2012*).

The UPR is a network of evolutionarily conserved signal transduction pathways that monitors the conditions in the ER lumen to induce a transcriptional response. In metazoan cells, three ER-resident transmembrane sensors, Ire1, PERK, ATF6 transmit information into the cytosol. Each sensor activates transcription factors that collaborate to drive expression of UPR target genes (*Walter and Ron, 2011*), including genes encoding ER-lumenal chaperones, such as BiP, an abundantly expressed Hsp70 family member.

Ire1 is a bifunctional transmembrane kinase/endoribonuclease that controls expression of the transcription factor XBP1 by a non-conventional splicing of its mRNA. Ire1 uses its ER-lumenal domain to detect unfolded proteins, and in response activates by homo-oligomerization, *trans*-autophosphorylation, and allosteric activation of its cytosolic nuclease modality (*Korennykh et al., 2009*; *Gardner and*

**eLife digest** Protein folding—the process by which a sequence of amino acids adopts the precise shape that is needed to perform a specific biological function—is one of the most important processes in all of biology. Any sequence of amino acids has the potential to fold into a large number of different shapes, and misfolded proteins can lead to toxicity and other problems. For example, all cells rely on signaling proteins in the membranes that enclose them to monitor their environment so that they can adapt to changing conditions and, in multicellular organisms, communicate with neighboring cells: without properly folded signaling proteins, chaos would ensue. Moreover, many diseases—including diabetes, cancer, viral infection and neurodegenerative disease—have been linked to protein folding processes. It is not surprising, therefore, that cells have evolved elaborate mechanisms to exert exquisite quality control over protein folding.

One of these mechanisms, called the unfolded protein response (UPR), operates in a compartment within the cell known as the endoplasmic reticulum (ER). The ER is a labyrinthine network of tubes and sacs within all eukaryotic cells, and most proteins destined for the cell surface or outside the cell adopt their properly folded shapes within this compartment. If the ER does not have enough capacity to fold all of the proteins that are delivered there, the UPR switches on to increase the protein folding capacity, to expand the surface area and volume of the compartment, and to degrade misfolded proteins. If the UPR cannot adequately adjust the folding capacity of the ER to meet the demands of the cell, the UPR triggers a program that kills the cell to prevent putting the whole organism at risk.

Researchers have identified the cellular components that monitor the protein folding conditions inside the ER. All eukaryotic cells, from unicellular yeasts to mammalian cells, contain a highly conserved protein-folding sensor called Ire1. In all species analyzed to date, Ire1 is known to activate the UPR through an messenger RNA (mRNA) splicing mechanism. This splicing event provides the switch that drives a gene expression program in which the production of ER components is increased to boost the protein folding capacity of the compartment.

Kimmig, Diaz *et al.* now report the first instance of an organism in which the UPR does not involve mRNA splicing or the initiation of a gene expression program. Rather, the yeast *Schizosaccharomyces pombe* utilizes Ire1 to an entirely different end. The authors find that *the* activation of Ire1 in *S. pombe* leads to the selective *decay* of a specific class of mRNAs that all encode proteins entering the ER. Thus, rather than increasing the protein folding capacity of the ER when faced with an increased protein folding load, *S. pombe* cells correct the imbalance by decreasing the load.

The authors also show that a lone mRNA—the mRNA that encodes the molecular chaperone BiP, which is one of the major protein-folding components in the ER—uniquely escapes this decay. Rather than being degraded, Ire1 truncates BiP mRNA and renders it more stable. By studying the UPR in a divergent organism, the authors shed new light on the evolution of a universally important process and illustrate how conserved machinery has been repurposed.

*Walter, 2011*). Activated Ire1 cleaves the *XBP1* mRNA at two discrete stem-loop structures, excising a short intron. The two severed exons are then ligated to produce spliced *XBP1* mRNA, which because of a frame-shift induced by the splicing event, are translated to produce active XBP1 (*Yoshida et al., 2001*; *Calfon et al., 2002*).

Ire1 was first discovered in the budding yeast *S. cerevisiae*, where it constitutes the core machinery of the cells' only UPR signaling pathway (*Cox et al., 1993*; *Mori et al., 1993*). *S. cerevisiae* Ire1 splices *Hac1* mRNA, encoding the yeast ortholog of XBP1, by a mechanism that was later found conserved in all metazoan cells (*Cox and Walter, 1996*; *Sidrauski et al., 1996*). Ire1-mediated mRNA splicing therefore is considered to be the most evolutionary ancient branch of the UPR.

By first approximation, the three UPR branches collaborate to effect comprehensive transcriptional outputs, thereby enhancing the capacity of the ER according to need. PERK superimposes another layer of control by reducing the load of proteins entering the ER through translational control (*Pavitt and Ron, 2012*). Similarly, Ire1 is thought to play a dual role in UPR regulation. In particular, *Hollien and Weissman (2006)* first discovered in *Drosophila* cells that Ire1 induction not only results in splicing

of *XBP1* mRNA but also mediates enhanced mRNA breakdown. This output of Ire1 activation, termed 'regulated Ire1-dependent decay' (RIDD), is conserved in mammalian cells, but not in *S. cerevisiae*, where transcriptional control via *Hac1* mRNA splicing remains the only known route of UPR signaling (*Niwa et al., 2005*; *Han et al., 2009*; *Hollien et al., 2009*). All identified RIDD target mRNAs are translated by membrane-bound ribosomes at the ER surface, where they are cleaved, most likely by Ire1 directly (*Han et al., 2009*; *Hollien et al., 2009*; *Cross et al., 2012*). Once nicked and no longer protected by their polyA tails and 5′ caps, mRNA fragments are quickly degraded by the RNA surveillance machinery (*Hollien and Weissman, 2006*; *Garneau et al., 2007*).

By contrast to the strictly conserved stem/loop structures found at *Hac1/XBP1* mRNA splice sites (*Gonzalez et al., 1999*), RIDD target mRNAs do not contain easily recognizable features in common. Consequently, RIDD is thought to arise by a more promiscuous cleavage mode of Ire1. It is unclear whether RIDD is mediated by an alternate conformation of activated Ire1, or whether it arises in a specific Ire1 oligomerization state, as high-order oligomerization may serve to locally enhance low affinity interactions through avidity effects. RIDD cleavage reactions have been reconstituted in vitro with recombinantly expressed purified Ire1, lending support to the notion that Ire1's endoribonuclease actvity, rather than another enzyme recruited to it, carries out the initial cleavage reaction (*Lee et al., 2011*; *Cross et al., 2012*).

Because of Ire1's dual output, the physiological consequences of RIDD have been difficult to decipher. RIDD has been suggested to play cytoprotective roles, such as contributing to important feedback control on proinsulin expression in pancreatic beta-cells or protecting liver cells from acetaminophen toxicity by degrading the mRNAs encoding the cytochrome P450 variants responsible for the drug's toxification (*Lipson et al., 2008*; *Hur et al., 2012*). RIDD has also been suggested to play cytotoxic roles as a major contributor driving cells into apoptosis after prolonged and unmitigated exposure to ER stress (*Han et al., 2009*).

Surprisingly, in the work presented here we found no evidence that Ire1 controls transcription in the UPR of *Schizosaccharomyces pombe*. Instead, in *S. pombe* Ire1 maintains ER homeostasis through two post-transcriptional mechanisms: it initiates RIDD of a large, select set of ER-targeted mRNAs and processes *Bip1* mRNA in an unprecedented way, thereby stabilizing it. Our studies reveal an unforeseen evolutionary plasticity in maintaining ER homeostasis.

## Results

### A functional UPR in fission yeast

UPR induction in all eukaryotic cells analyzed to date involves the Ire1-mediated, non-conventional splicing of *Hac1/XBP1* mRNA. The splice sites at which Ire1 cleaves the mRNA to initiate splicing lie in well-conserved stem/loop structures that are readily identified (*Gonzalez et al., 1999*). We and others were therefore perplexed when bioinformatic analyses failed to identify Hac1/XBP1 orthologs in *S. pombe* and other yeasts of the same genus (*Figure 1a*) (*Hooks and Griffiths-Jones, 2011*; *Frost et al., 2012*). The Hac1/XBP1 transcription factors are well conserved between species and are easily recognized by sequence alignment among the superfamily of bZIP transcription factors (*Figure 1—figure supplement 1*). By contrast, Ire1 is well conserved in *S. pombe*, with all of the functionally important hallmarks identified in other eukaryotes, including its ER lumenal unfolded protein sensing domain and its cytosolic kinase and RNase domains. Moreover, Ire1 was essential for *S. pombe* growth on tunicamycin (Tm) (*Figure 1b*), which induces ER stress by blocking *N*-linked glycosylation, indicating that Ire1 serves an essential function in allowing cells to cope with ER stress. This function required Ire1's RNase activity, as *Ire1(H1018N)* carrying a single amino acid substitution of a catalytic residue in Ire1's RNase active site failed to support cell growth on tunicamycin (*Figure 1b*).

### ER stress dependent mRNA down-regulation

To address the conundrum posed by the missing Ire1 splicing substrate in *S. pombe*, we first explored the scope of UPR-dependent changes in gene expression. To this end, we isolated polyA+ RNA from wild type and *Ire1Δ* cells, in which the UPR was induced with the reducing agent dithiothreitol (DTT). DTT causes ER stress by impairing disulfide bond formation in the ER. The purified mRNA population was reverse-transcribed and subjected to deep-sequencing. Unexpectedly, we observed widespread Ire1-dependent mRNA down-regulation, but virtually no mRNA up-regulation (*Figure 1c*). Thirty-nine mRNA species were reduced by more than twofold in a DTT- and Ire1-dependent manner (*Figure 1c*,

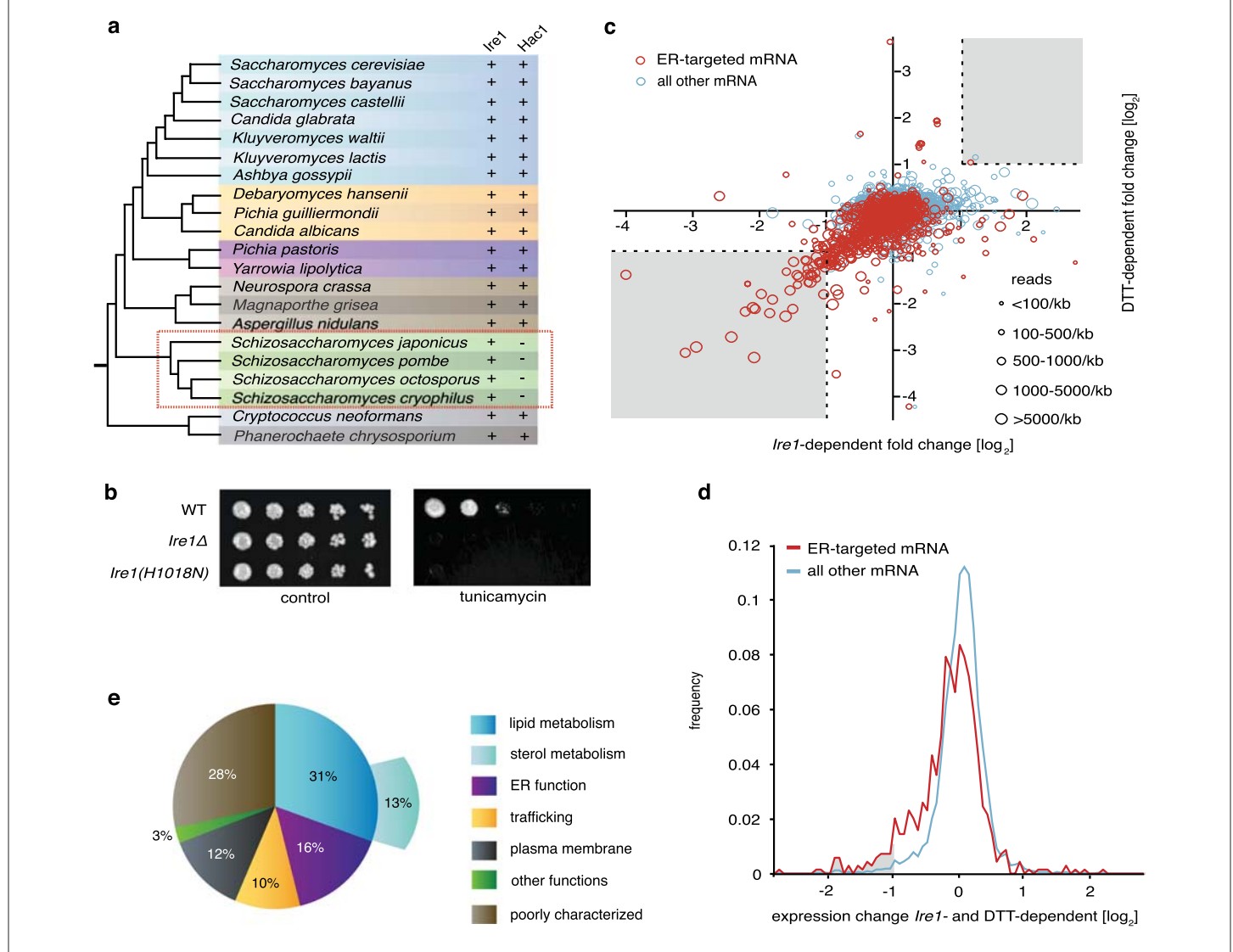

**Figure 1**. The UPR in fission selectively down-regulates ER-targeted mRNAs. (**a**) Phylogenetic tree showing the components of the UPR in yeasts. The presence of recognizable orthologs of Ire1 and Hac1 is indicated. (**b**) Viability assay by serial dilution of wild type, *Ire1Δ* and *Ire1(H1018N)* cells spotted on solid media with or without 0.03 μg/ml of the ER stress inducer tunicamycin (Tm). Plates were photographed after 3 day of growth at 30°C. (**c**) Strand-specific polyA+ enriched mRNA-Seq analysis of annotated ORFs. The plot indicates the fold change (log2) of transcript abundance in DTT-stressed *Ire1Δ* cells (2 mM DTT, 1 hr) compared to DTT-stressed wild type cells (2 mM DTT, 1 hr) in the x-axis, and transcript abundance in unstressed wild type cells compared to DTT-stressed wild type cells (2 mM DTT, 1 hr) in the y-axis. Symbol sizes indicate abundance classes for each mRNA (reads per kilobase). Transcripts encoding proteins with a signal sequence or transmembrane segment are colored red, all other transcripts are colored blue (Figure 1—source data). (**d**) DTT-dependent and Ire1-dependent expression changes of transcripts displaying a signal sequence or a transmembrane domain. The skew of the left tail of the distribution indicates an enrichment (p<1×10−20) of down-regulated mRNAs. Coloring is as in *Figure 1d*. (**e**) Distribution of gene-ontology (GO) annotations for Ire1-dependent down-regulated mRNAs. Percentages indicate genes within a particular GO category in relation to the total number of genes that have a GO annotation (N=39) (see *Figure 1—figure supplement 3* for annotated list of genes).

The following source and figure supplements are available for figure 1:

**Source data 1**. Gene expression and fold change
**Figure supplement 1**. Alignment of DNA binding domain of Hac1 (bZIP) homologues in different yeast species.
**Figure supplement 2**. Plot depicting ER-targeted mRNAs abundance [log10] (reads per million) versus DTT-dependent expression changes [log2] for wild type cells.
**Figure supplement 3**. Ontology of genes down-regulated more than twofold Ire1-and DTT-dependent.

bottom left grayed area). Most members of this set of down-regulated mRNAs were abundantly expressed, as depicted by the size of the plotted circles. Down-regulation, however, did not correlate with mRNA abundance (*Figure 1—figure supplement 2*). Intriguingly, the set of down-regulated genes exclusively encoded proteins targeted to the ER (identified by signal sequences and/or transmembrane segments) (*Figure 1c*, red circles). As shown in *Figure 1d*, the genome-wide profile of Ire1- and ER stress-dependent mRNA changes of genes encoding ER-targeted proteins is skewed to a significantly greater extent toward down-regulation than that of other mRNAs (p<1×10⁻²⁰). More than half of the most down-regulated mRNAs encode proteins with annotated functions in the secretory pathway, in particular proteins involved in lipid metabolism, trafficking, and ER functions (*Figure 1e*).

As the reduction in mRNA abundance was ER stress- and Ire1-dependent, we next explored if Ire1 could be directly involved in destabilizing ER-bound mRNAs. To this end, we sought to trap any putative primary Ire1-cleavage products prior to degradation by deleting *Ski2*, which encodes a helicase component of the cytosolic Ski complex (cytosolic exosome) that mediates 3′ → 5′ RNA decay. Northern blot analysis of *Ski2Δ* cells revealed that *Gas2* mRNA (which is down-regulated 2.5-fold in an ER stress and Ire1-dependent manner) yielded two discrete cleavage products upon ER stress (*Figure 2a*). *Gas2* mRNA cleavage was dependent on Ire1, as no mRNA reduction and no cleavage products were observed in *Ire1Δ Ski2Δ* double deletion cells (*Figure 2a*). Another target, *Yop1*, behaved similarly (*Figure 2—figure supplement 1*). In time-course experiments, reduction of *Gas2* mRNA and accumulation of the cleavage products peaked at 30 min after UPR induction (*Figure 2b*); at later time points the abundance of intact full-length mRNA increased, suggesting that newly transcribed mRNA is not cleaved if the Ire1-dependent cleavage products are not further degraded. Indeed, *Ski2Δ* cells failed to grow on plates containing tunicamycin (*Figure 2c*), indicating that an intact mRNA decay is important for *S. pombe* cells to cope with ER stress.

To determine the mRNA cleavage sites genome-wide, we prepared total RNA fractions from *Ski2Δ* and from *Ire1Δ Ski2Δ* cells. We used tRNA ligase to attach linker sequences specifically to those RNA fragments terminating in a 2′,3′-cyclic phosphate, which is the expected product of Ire1-catalyzed RNA cleavage (*Schutz et al., 2010*). We then amplified the cleavage products in 3′ RACE reactions priming at the linker sequence. Alignment of the sequencing data to the *S. pombe* genome identified the 3′ ends of Ire1-dependent fragments. In particular, we identified 39 Ire1-dependent fragments mapping to 24 of the most down-regulated genes, as shown in *Figure 2d* for *Gas2* mRNA (left panel). By size estimation, the major Ire1-dependent peak corresponded to the smaller, more abundant *Gas2* mRNA cleavage product (labeled ▲ in *Figures 2a and 2b*). A second, less abundant short fragment was also observed in the sequencing data (labeled ✕ in *Figure 2d*). (Fragment ✕ was absent or below the detection limit on the Northern blot unless the primary cleavage site was mutated (see *Figure 2g*, discussed below).) Spliceosomal U6 RNA normally terminates in a 2′,3′-cyclic phosphate and thus provided a valuable control for the ligation reaction (*Figure 2d*, right panel).

Alignment of the experimentally determined Ire1-dependent cleavage sites revealed a core motif with a signature of three conserved nucleotides (UG\C) that flank the Ire1-dependent cleavage sites at positions −2, −1, and +1 with an additional strong bias against G in position +2 (*Figure 2e*). Most mapped mRNA cleavage sites (34 of 39), including those in *Gas2* mRNA, localized within the open reading frames. Indeed, a *Gas2* reporter construct transcribed off a heterologous alpha-tubulin (*Nda2*) promoter and containing only the *Gas2* ORF flanked by heterologous 5′ and 3′ tubulin untranslated regions (UTRs), was down-regulated upon ER stress in an Ire1-dependent manner (*Figure 2f*). This degradation was quantitatively comparable to that of the native *Gas2* transcript (*Figure 2—figure supplement 2*), indicating that the information contained within the *Gas2* ORF is sufficient to confer susceptibility to Ire1-dependent cleavage.

To assess the functional importance of the identified *Gas2* mRNA cleavage site experimentally, we mutated the UGC-residues of the ▲-site (UG\CU). As expected, ER stress-dependent cleavage of the *Gas2* reporter mRNA at the mapped site was abolished (*Figure 2g*). In its place, however, we observed two new Ire1-dependent fragments (labeled ✕ and ✱). Scanning gel densitometry revealed that fragment ✕ is distinctly smaller than fragment ▲, and hence represents a cryptic site that is only utilized when site ▲ is mutated. Fragment ✱ likely corresponds to the lower abundance cleavage product observed in *Figures 2a and 2b*, which becomes more prominent in the mutant construct. Taken together, we conclude that Ire1-dependent mRNA cleavage in *S. pombe* is sequence dependent.

The data presented so far suggest that homeostatic control of ER protein folding is regulated differently in *S. pombe* than *S. cerevisiae*. Rather than relying on a transcriptional program to upregulate

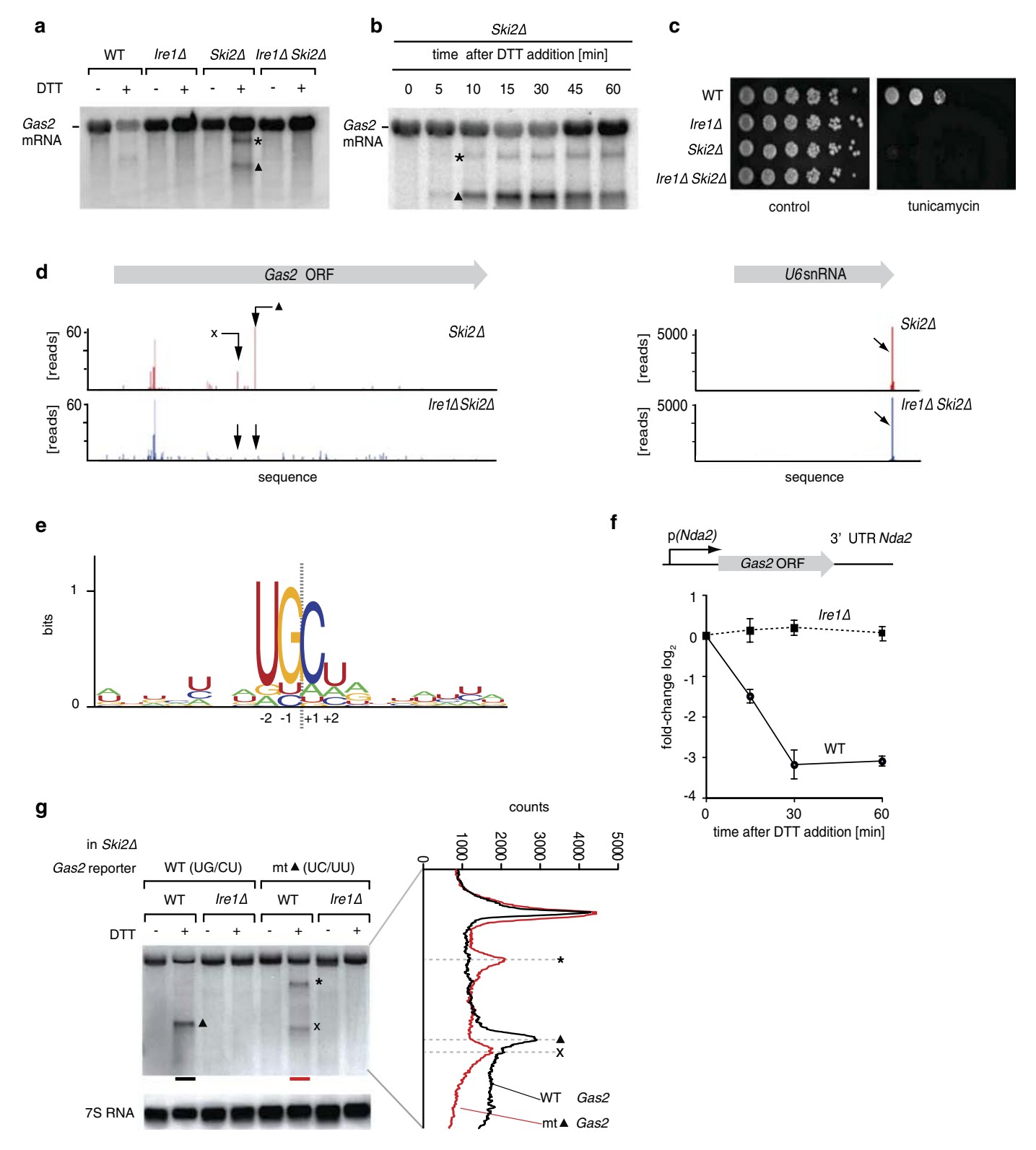

**Figure 2**. Ire1 cleaves down-regulated mRNAs at specific sequences. (**a**) Northern blot of total RNA extracted from wild type, *Ire1Δ*, *Ski2Δ* and double mutant ER stressed *Ire1Δ Ski2Δ* cells (2 mM DTT, 1 hr). A probe complementary to the 5′ UTR of *Gas2* was used to detect cleavage products. The triangle and asterisk indicate two different mRNA cleavage products. (**b**) Northern blot of total RNA extracted from ER-stressed *Ski2Δ* cells (2 mM DTT). (**c**) Viability assay by serial dilution of wild type, *Ire1Δ*, *Ski2Δ and Ire1Δ Ski2Δ* cells spotted on solid media with or without ER stress as in **Figure 1b**. (**d**) *Figure 2. Continued on next page*

*Figure 2. Continued*

RNA-sequence read density map of the *Gas2* locus derived from 3′ end deep-sequencing data. Library was generated by ligating a DNA-linker using tRNA ligase to 3′ end mRNAs with 2′,3′-cyclic phosphates in *Ski2Δ* and *Ire1Δ Ski2Δ* ER-stressed cells (2 mM DTT, 30 min). The arrows indicate two Ire1-depedent cleavage sites. The U6 snRNA locus was used as a positive control (Figure 2—source data). (**e**) Ire1 RNA sequence recognition motifs generated by deep-sequencing analysis of tRNA ligase-generated RNA libraries of 39 mRNA targets down-regulated twofold or more in an Ire1-dependent manner. The resulting position weight matrices are illustrated as a logo. The dotted line indicates the cleavage site. (**f**) Real-time qPCR of a chromosomally integrated reporter containing the coding sequence of *Gas2* under the control of the *Nda2* (tubulin) promoter and including the UTRs of *Nda2*. A time course after DTT addition (2 mM) is shown. Endogenous *Nda2* was used as a normalization control. Error bars: standard deviation. (**g**) Northern blot analysis of total RNA extracted from *Ski2Δ* and *Ire1Δ Ski2Δ* cells carrying a mutant version of the reporter indicated in (f) were the putative Ire1 cleavage site (▲, UG\CU→ UC\UU) was mutated. Note that the band labeled ✖ migrates distinctly faster, as shown by scan on the right.

The following source and figure supplements are available for figure 2:

**Source data 1**. 2′,3′ cyclic-phosphate 3′ end mapping

**Figure supplement 1**. Northern blot of total RNA isolated from wild type, *Ire1Δ*, *Ski2Δ* and double mutant *Ire1Δ Ski2Δ* ER stressed cells (2 mM DTT, 1 hr).

**Figure supplement 2**. Comparison of ER stress-dependent down-regulation of endogenous *Gas2* mRNA (2 mM DTT, 1 hr: deep-sequencing) and reporter *Gas2* mRNA (from qPCR: 2 mM DTT, 1 hr; see also ***Figure 2f***).

genes that enhance ER protein folding capacity as in *S. cerevisiae*, *S. pombe* cells reduce the amount of specific proteins entering the organelle by decreasing the level of ER-targeted mRNAs using Ire1-dependent mRNA degradation.

## *Bip1* mRNA processing and stabilization in response to ER stress

In all species analyzed to date, Bip1 is a major UPR target gene that is upregulated when cells experience ER stress. Paradoxically, we found *S. pombe Bip1* mRNA among the 39 down-regulated mRNAs identified by the analyses shown in ***Figure 1c***. Analysis by Northern blotting yielded seemingly conflicting results: by this analysis, *Bip1* mRNA was fourfold more abundant in ER stressed cells (***Figure 3a***). Intriguingly, the appearance of a faster migrating mRNA species ('*tBip1* mRNA') indicates that *Bip1* mRNA changes size in cells experiencing ER stress (***Figure 3a***, lanes 3–4). Appearance of the *tBip1* mRNA species was Ire1-dependent and in wild type cells accounted for the increase in overall mRNA abundance. The increase did not result from augmented transcription. We measured the activity of a heterologous reporter in which the *Bip1* promoter was fused to GFP and showed no Ire1-dependent change in mRNA abundance with ER stress (***Figure 3b***). In agreement with this result, we found that the stability of an mRNA bearing the *Bip1* ORF and 3′ UTR showed a more than threefold increase in half-life from $T_{1/2}$=20 min for the unprocessed form present in unstressed cells to $T_{1/2}$=70 min for the processed form present in ER-stressed cells (***Figure 3c***). Furthermore, the *Bip1* 3′ UTR and the presence of a signal sequence were sufficient to a heterologous mRNA construct to confer Ire1-dependent processing (***Figure 3— figure supplement 1***).

Sequencing of the expressed genome in UPR-induced and uninduced cells revealed the molecular difference between *Bip1* and *tBip1* mRNA (***Figure 3d***). For these experiments, we extracted total RNA and then, without selecting for polyA$^+$ RNA, removed rRNA by subtractive hybridization. After reverse transcription, deep-sequencing of the cDNA pool from uninduced cells revealed good coverage of reads spanning the entire *Bip1* mRNA including its 5′ and 3′ UTR (***Figure 3d***, left, blue profile). By contrast, cDNA isolated from DTT-treated cells revealed a precipitous drop in reads mapping to the 3′ end (***Figure 3d***, left, red profile and ***Figure 3—figure supplement 2***). We also performed 3′-RACE to determine the 3′ end of *tBip1* mRNA. The sequence of the amplified DNA confirmed that *tBip1* mRNA lacks a polyA tail and terminates at G373 in the 3′ UTR (***Figure 3d***, right panel). In seven independently isolated clones, we found no sequence variations in the *tBip1* linker junction. The sequences flanking G373 align with the UG\CU motif (***Figure 2e***), suggesting that *tBip1* mRNA is produced by truncation of *Bip1* mRNA in an Ire1-dependent RNA cleavage reaction that resembles those of the Ire1-dependently down-regulated mRNAs described above.

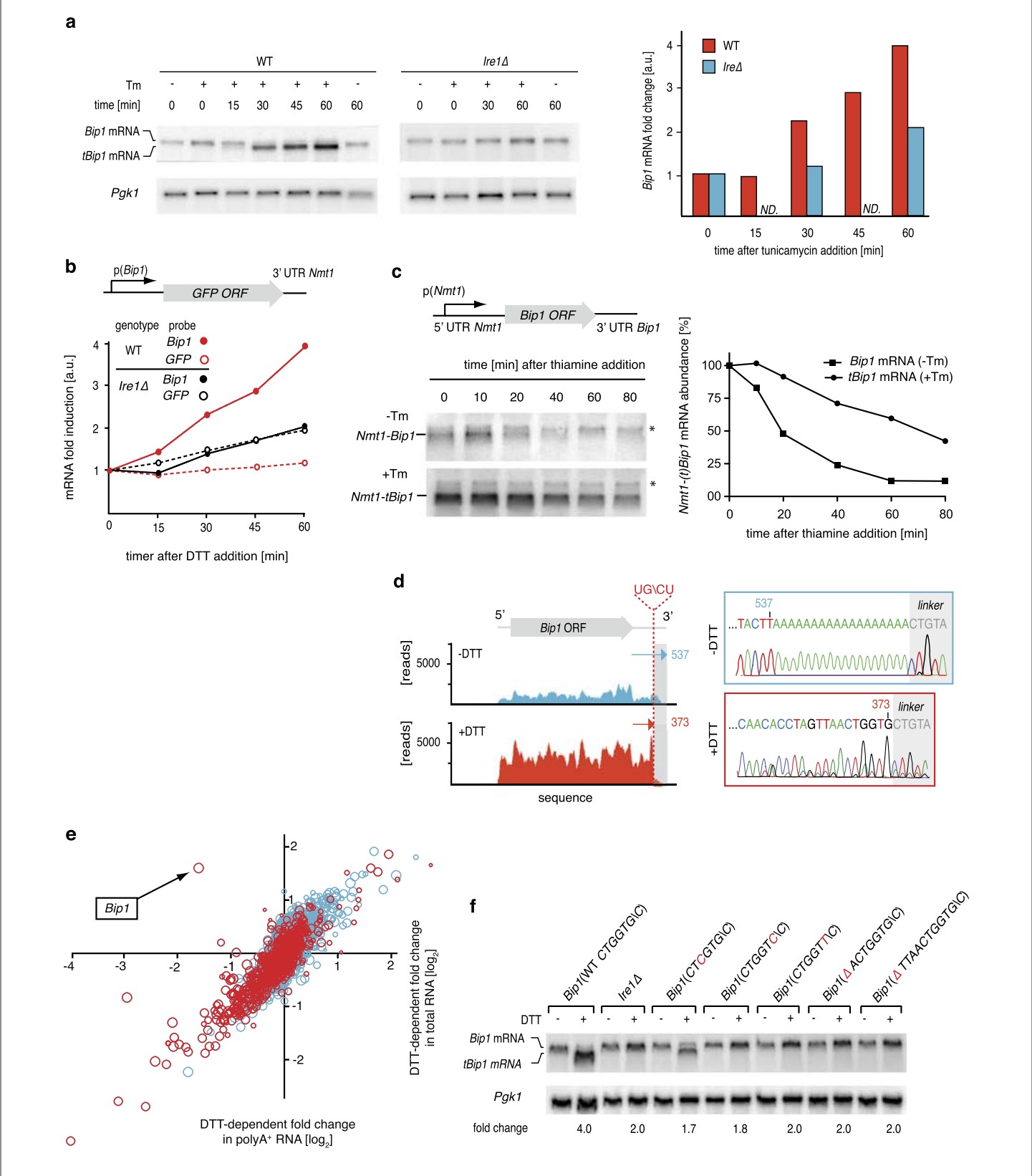

**Figure 3**. Ire1 truncates *Bip1* mRNA within the 3' UTR. (**a**) Northern blot analysis of total RNA extracted from wild type and *Ire1Δ* cells untreated or treated with tunicamycin (1 µg/ml), and hybridized with a probe complementary to the ORF of *Bip1* mRNA. Right panel: quantitation normalized

*Figure 3. Continued on next page*

*Figure 3. Continued*

to *Pgk1* mRNA. (**b**) The abundance of a *GFP* mRNA driven by the *Bip1* promoter (black) compared to endogenous *Bip1* mRNA was determined as a time course after DTT (2 mM) addition by quantitative Northern blotting. (**c**) Wild type cells bearing a construct encoding the *Nmt1* 5′ UTR, *Bip1* ORF and *Bip1* 3′ UTR driven by the *Nmt1* promoter were pre-treated with tunicamycin (0.25 μg/ml, 1 hr). At different time points after thiamine (15 μM) addition (to effect transcriptional shut-off of the *Nmt1* promoter), RNA was extracted and analyzed by Northern hybridization. Blots were probed for the *Nmt1* 5′ UTR. *Nmt1-Bip1* mRNA and *Nmt-tBip1* mRNA were quantitated and normalized to the unspecific band (asterisk). (**d**) RNA-sequence read density map of the *Bip1* locus derived from mRNA-enriched (ribosome depleted) RNA in wild type cells untreated or treated with DTT (2 mM DTT, 1 hr; left panels). Data are representative of one of two biological replicates. Single nucleotide resolution of the 3′ terminus of *Bip1* mRNA determined by 3′ RACE (right panels). (**e**) Mutational analysis of the *Bip1* mRNA cleavage site by Northern blotting. Total RNA was extracted from wild type, *Ire1Δ* or cells carrying mutations in the *Bip1* 3′ UTR mRNA. Cells were treated with 2 mM DTT, 1 hr or left untreated as indicated. The fold-changes indicate *Bip1* mRNA abundance relative to that of *Pgk1* mRNA. (**f**) Strand-specific, mRNA enriched (after removal of ribosomal RNA) deep-sequence analysis of annotated ORFs (y-axis) compared to strand-specific polyA+ enriched mRNA deep-sequence analysis of annotated ORFs (x-axis) (see Figure 1—source data). The plot indicates the ratio of transcript abundance in unstressed vs DTT-stressed (2 mM DTT, 1 hr) wild type cells. Symbol sizes and colors are as described in *Figure 1c*.

The following figure supplements are available for figure 3:

**Figure supplement 1**. Northern blot analysis of wild type cells bearing a construct expressing a fusion protein of GFP preceded by the Bip1 signal sequence.

**Figure supplement 2**. Sequencing read coverage of the 3′; end nucleotide positions in tBip1 mRNA from derived from mRNA enriched by subtractive hybridization against rRNA (Ribominus kit, Invitrogen kit).

Mutational analysis of the cleavage site confirmed that specific sequences are required. Mutation of G373 to C or U and its deletion together with preceding nucleotides abolished Ire1-dependent *Bip1* mRNA processing (*Figure 3f*). By contrast, a mutation of the preceding G370 to C diminished cleavage only marginally (less than twofold). In all analyzed mutants of *Bip1* mRNA, UPR-induction increased abundance of the transcript approximately twofold (a level comparable to that observed in *Ire1Δ* cells) (*Figure 3a*, right panel), whether processing took place or not, perhaps due to compensatory transcriptional regulation that is independent of Ire1. For all mutants, however, the increased abundance stayed shy of the fourfold increase observed for wild type *Bip1* mRNA.

As *Bip1* mRNA truncation resulted in a loss of the polyA tail, this result resolves the paradox of why *Bip1* mRNA appeared to be down-regulated in the polyA+ mRNA pool analyzed in *Figure 1c*. Indeed, directly comparing the UPR-dependent fold-change in mRNA abundance of polyA+ RNA and rRNA-depleted total RNA uniquely positioned *Bip1* sequences as an anti-correlated outlier, whereas all other mRNAs were well correlated between the samples (*Figure 3e*). From these data we conclude that, remarkably, *Bip1* mRNA is the only stable mRNA in the cell that loses its polyA tail upon UPR induction.

It was unexpected to find an mRNA that had lost its polyA tail to be more stable in cells.

To determine the translation proficiency of *tBip1* mRNA, we subjected UPR-induced cells to polysome profiling. These experiments confirmed that despite lacking its polyA tail, *tBip1* mRNA sedimented in the polyribosome fractions in sucrose gradients (*Figure 4a*). Moreover, ribosome footprinting demonstrated that *Bip1* mRNA in uninduced cells and *tBip1* mRNA in UPR-induced cells were engaged with actively translating ribosomes, mapping throughout the *Bip1* ORF (*Figure 4b*). The larger number of reads obtained upon UPR induction correlated with the higher abundance of *tBip1* mRNA. We note a significant ribosome occupancy preceding the translation start site in both *Bip1* and *tBip1* mRNA most likely presenting previously unrecognized small uORFs (see *Figure 4—figure supplement 1* for a zoomed-in view). The relative ribosome occupancy of these putative uORFs did not change with UPR induction. Translation of the processed mRNA resulted in an enhanced steady-state concentration of Bip1 protein, as shown by quantitative Western blotting (*Figure 4—figure supplement 2*).

To assess the physiological consequences of this unique regulatory mechanism of Bip1 expression, we explored the growth of strains carrying a mutation of the *Bip1* mRNA processing site (ΔTTAACTGGTG\C). Liquid cultures of *Bip1* mRNA mutant, that were exposed to a pulse of ER stress (tunicamycin) and allowed to recover after washout of the drug showed a marked growth delay in early

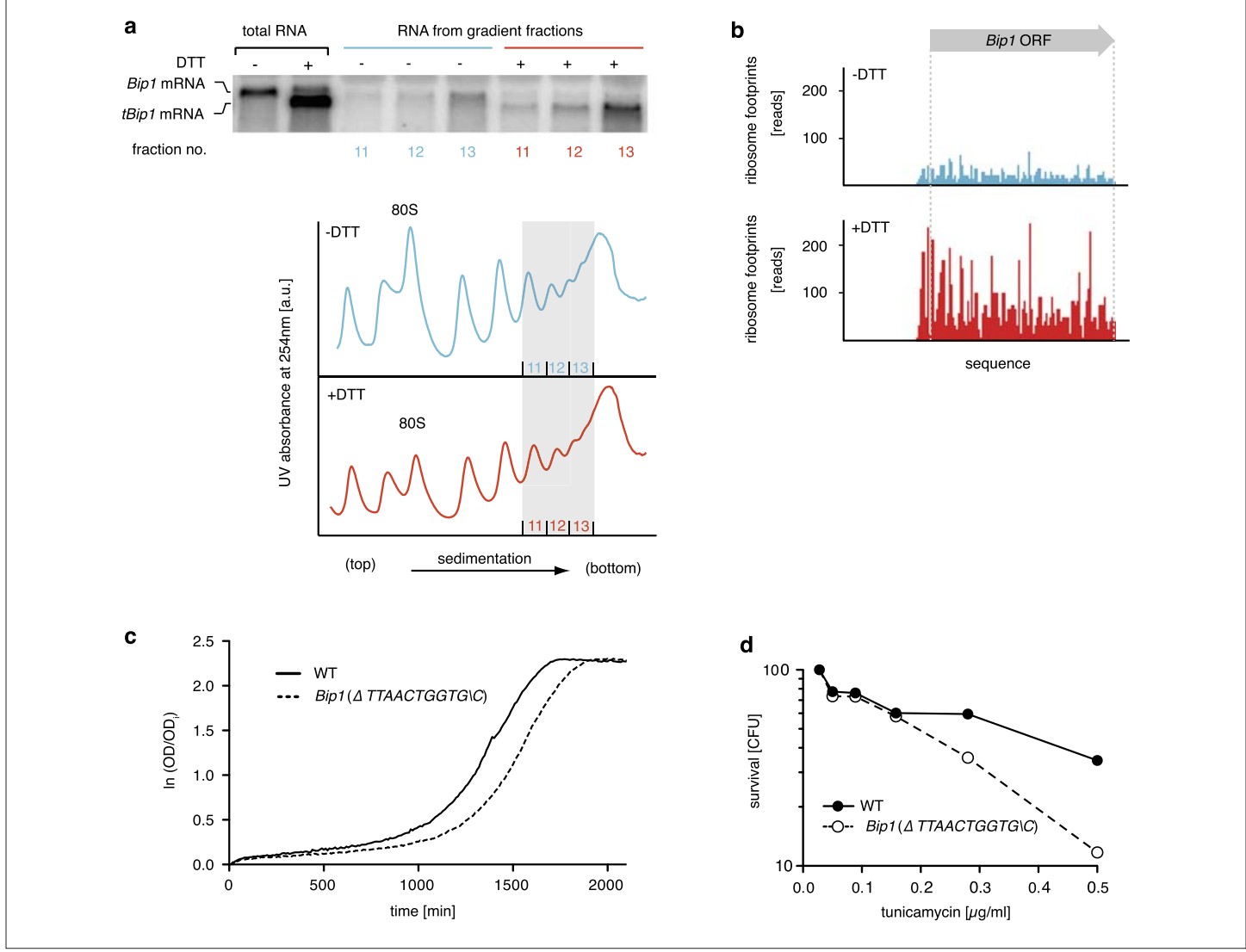

**Figure 4**. *tBip1* mRNA is translated and is important for fitness during ER stress. (**a**) Northern blot analysis of the distribution of total or *tBip1* mRNA in polyribosomes from extracts of unstressed or ER-stressed (2 mM DTT, 1 hr) cells. Fractions 11, 12, 13 of the sucrose gradients (lower panels) were analyzed by Northern blotting (upper panels). (**b**) Ribosome footprints (as described in *Materials and methods*) of *Bip1* mRNA in unstressed or ER stressed cells. The region that depicts ribosome density preceding the *Bip1* ORF is shown enlarged in **Figure 4—figure supplement 1** (Figure 4—source data 1). (**c**) Cell growth of wild type cells and cells carrying a deletion of *Bip1* mRNA cleavage sites (*Bip1*(Δ *TTAACTGGTGC*)). Cells were treated with tunicamycin (0.5 μg/ml) for 3 hr and then recovered from ER stress by washing out the drug and re-seeding in warm fresh media. Optical density (OD) at 660 nm was measured immediately afterward in 10 min intervals. (**d**) Viability assay of the same cells as in (c). The percentage of viable cells was determined by counting the number of colony-forming units (CFU) after growth for 3 hr at varying tunicamycin concentration.

The following source and figure supplements are available for figure 4:

**Source data 1**. Ribosome footprint reads for Bip1

**Figure supplement 1**. Zoomed-in ribosome occupancy profile of Bip1 mRNA around the start AUG codon of wild type cells. Putative non-canonical uORFs are highlighted in both *Bip1* and *tBip1* mRNA derived from untreated and DTT-treated (2 mM, 1 hr) cells.

**Figure supplement 2**. Upper panel: Western blot of Bip1 and, as a loading control, RNA polymerase II CTD repeat (RNAPII). Wild type and *Ire1Δ* cells were treated with tunicamycin 0.5 μg/ml and samples were taken at indicated time points. Lower panel: Quantification of Western blotting with values normalized to RNAPII.

**Figure supplement 3**. Viability assay by serial dilution of wild type, *Ire1Δ* and different Bip1 mRNA cleavage mutants spotted on solid media with or without 0.03 μg/ml of the ER stress inducer tunicamycin. Plates were photographed after 3 day of growth at 30°C.

log phase (*Figure 4c*) and enhanced cell death (*Figure 4d*), indicating that *Bip1* mRNA processing is important for maintaining cell fitness in the face of ER stress. By contrast to cell growth in liquid culture, *Bip1* mutant cells grew on UPR-inducing tunicamycin plates only marginally worse that wild type cells (*Figure 4—figure supplement 3*). The importance of *Bip1* mRNA processing, therefore, varies with growth conditions.

## Discussion

We have begun to characterize the UPR in fission yeast. To our surprise, we discovered that—by contrast to all other eukaryotes studied to date—*S. pombe* does not utilize Ire1, the most ancient ER stress sensor, to control transcription. Rather, *S. pombe* exclusively relies on two means of Ire1-dependent *post*-transcriptional regulation to cope with ER stress:

  i. Regulated Ire1-dependent mRNA decay (RIDD) of a large and highly select set of mRNAs, all of which are predicted to be translated by membrane-bound ribosomes at the ER. The mechanistic features of mRNA degradation in *S. pombe* are highly reminiscent of RIDD previously described in insect and mammalian cells, where it accompanies Ire1-dependent mRNA splicing.
  ii. Processing of *Bip1* mRNA within its 3' UTR, leading to loss of its polyA tail, which—counter-intuitively—results in its stabilization. The mechanistic features of this unprecedented mRNA processing step resemble the initial Ire1-dependent nucleolytic step of RIDD but *Bip1* mRNA then escapes further decay.

Our results shine new light on how ER homeostasis can be maintained and underscore the fascinating divergence of solutions that different species evolved to achieve this task.

### The physiological role of RIDD in *S. pombe*

ER stress arises from insufficiencies in handling the protein-folding load in the ER lumen. Homeostasis, therefore, can be reestablished in two principal ways: increasing the capacity of the ER to handle the load—or decreasing the load to meet the capacity. Here we show that mRNA decay can serve as the sole means of resolving ER stress without transcriptional up-regulation of classical UPR target genes. The identified transcripts targeted for RIDD compose a subset of mRNAs, all encoding proteins that reside in or traverse the secretory pathway. Being translated by membrane-bound ribosomes, these mRNAs are therefore in an appropriate cellular location to meet activated Ire1.

As all proteins encoded by RIDD target mRNAs enter the ER lumen, their synthesis by definition contributes to the burden of the ER protein folding machinery. RIDD therefore helps reduce the protein-folding load. It is less clear however whether such reduction would have a major impact. Indeed, a back-of-the-envelope calculation indicates that in *S. pombe* RIDD reduces the total protein influx into the ER by only 15%, even under the severe ER stress conditions explored here experimentally. This estimate derives from our ribosome footprinting data in normal vs ER-stressed cells: We scored the relative translational engagement of all mRNAs encoding proteins displaying signal sequences or transmembrane regions to estimate flux of newly synthesized polypeptides in to the ER and calculated the impact of RIDD on this set (see *Materials and methods*). It is difficult to envision how a mere 15% reduction of bulk protein flux into the ER would suffice to alleviate an otherwise lethal ER stress.

It is possible that RIDD preferentially targets proteins that are particularly difficult to fold and hence might have a disproportional impact on the protein folding load in the ER lumen. Indeed, Ire1 may be localized to the vicinity of mRNAs encoding such proteins by interactions of its ER lumenal unfolded protein sensing domain with portions of the nascent polypeptide chains that have entered the ER lumen, as previously proposed (*Hollien and Weissman, 2006*; *Ron, 2006*). An alternative and not mutually exclusive view poses that RIDD qualitatively changes the gene expression profile. In support of this notion, we notice that the population of RIDD target mRNAs is highly selective. mRNAs encoding proteins involved in lipid metabolism are highly enriched (31% of RIDD target mRNA as compared to 6.7% in all ER-targeted mRNAs). Moreover, we find that RIDD targets encoding proteins involved in sterol metabolism are particularly enriched (13% as compared to 1.3%). How reduced sterol synthesis would counteract the toxic effects of ER stress remains unclear. One

possible explanation would be that ER stress limits sterol exit through vesicular transport and a compensatory reduction in sterol synthesis becomes important to sustain basic ER functions, perhaps by maintaining appropriate membrane fluidity (*Nilsson et al., 2001*; *Feng et al., 2003*). In this way, RIDD (akin to other degradative pathways) could adjust basic metabolic parameters in the cell (*Bernasconi et al., 2012*).

## RNA cleavage and recognition

Previous work strongly suggests that Ire1 is the nuclease that initiates RIDD in metazoan cells (*Han et al., 2009*; *Hollien et al., 2009*; *Cross et al., 2012*). Our results provide two further lines of evidence in support of this view: first, we show that an Ire1 RNase active site mutant, *Ire1(H1018N)*, is unable to sustain cell growth on ER stress-inducing media. This mutation was designed to block catalysis while retaining hydrogen-bonding interactions of the amino acid side chain. Indeed, the equivalent single amino acid substitution in *S. cerevisiae* Ire1, Ire1(H1080N), reduces catalytic activity by >$10^5$-fold. Otherwise, Ire1(H1080N) is indistinguishable from wild type Ire1, both in its oligomerization and structural properties as determined by crystallography (*Korennykh et al., 2011*). Second, we showed that cleaved RIDD target mRNAs carry a 2′,3′-cyclic phosphate group, which is a prerequisite for the ligation reaction (tRNA ligase) used in the genome-wide mapping of mRNA ends created upon ER stress. Ire1 and tRNA endonuclease (and the Ire1-family member RNase L found in mammalian cells) are the only cytoplasmic nucleases known to produce such products.

In *S. cerevisiae*, Ire1 has a single known substrate, *Hac1* mRNA (*Niwa et al., 2005*). *Hac1* and *XBP1* mRNA have highly conserved and readily recognizable stem loop structures that demarcate the splice sites. Cleavage occurs at a universally conserved G always found at position 3 in the seven-base loop (*Gonzalez et al., 1999*). This information is interchangeable between species: constructs derived from *S. cerevisiae Hac1* mRNA are properly spliced in mammalian cells, and yeast Ire1 recognizes and precisely cleaves *XBP1* mRNA-derived substrates (*Niwa et al., 1999*). We show that RIDD target mRNAs contain a short three-base UGC consensus at the Ire1 cleavage site where cleavage occurs after the G, consistent with cleavage specificity previously assigned to Ire1 (*Gonzalez et al., 1999*).

Thus by contrast to our understanding of the RNA-elements directing Ire1-cleavage that initiates *Hac1* and *XBP1* mRNA splicing, the information that directs mRNAs into RIDD remains vastly underspecified. By chance, UGC triplets occur much more frequently in mRNAs than Ire1 cleavage sites. Therefore, the information that specifies an mRNA as an Ire1 substrate must require additional elements. Potential determinants may lie in sequence or secondary structure determinants that to date have escaped bioinformatics identification. We and others note that many of the identified cleavage sites lie in loops of potential hairpin structures (*Hur et al., 2012*; *Oikawa et al., 2010*); however, the position of the scissile G in the loops is not conserved. These structures therefore do not provide a structurally plausible explanation. Alternatively, Ire1's lumenal domain may become preferentially engaged with nascent polypeptide chains that display higher affinity and/or longer exposed peptide sequences, thereby selecting mRNAs co-translationally by recognizing features in the encoded protein. The concept of Ire1 recruitment, whether through interactions via the nascent chain or elements in the mRNA per se, is supported by our finding that mutation of one cleavage site (UG\C → UC\U) in *Gas2* mRNA gives rise to cleavage at alternative sites. These data indicate that local proximity rather than the RNA sequence surrounding the immediate cleavage site may guide substrate selection.

In order to stabilize the primary Ire1 cleavage products, we used mutant cells impaired in exosome function catalyzing 3′ → 5′ RNA degradation (*Anderson and Parker, 1998*; *Gatfield and Izaurralde, 2004*). Intriguingly, we observed that *Gas2* mRNA decay in *Ski2Δ* mutant strains was observed only transiently, peaking at 30 min after ER stress induction. This kinetic behavior suggests that clearance of Ire1-cleavage products by Ski2 may be a requirement for continued Ire1 activity. Thus, initiation of RIDD by Ire1 and further decay may be obligatorily coupled.

## Processing of *Bip1* mRNA

The observation that *S. pombe Bip1* mRNA changes size upon ER stress dates back to 1992 (*Pidoux and Armstrong, 1992*). To date, *Bip1* mRNA processing has not been observed in any other species. The phenotypic observation of the mRNA size shift has been deployed many times as an ER stress indicator (*D'Alessio et al., 1999*), but, surprisingly, its origin has not been investigated. As we suggest

here, *Bip1* mRNA is also cleaved by Ire1, yet escapes decay. The Ire1 cleavage site shares the same features described above for RIDD target mRNAs. During ER stress-induced processing, *Bip1* mRNA loses a portion of its 3′ UTR and polyA tail. The resulting processed *tBip1* mRNA is more stable and hence is present at an increased steady-state concentration. A plausible explanation for the increased stability of *tBip1* mRNA is the loss of an RNA degron located in the severed portion of the 3′ UTR. *tBip1* mRNA is actively translated with its ribosome density paralleling its increased abundance. Although polyA tails are generally linked to stability and translational efficiency, histone mRNAs, which likewise lack polyA tails, provide precedence for such an exception to the rule (*Marzluff et al., 2008*). Histone mRNAs terminate in a well-conserved 3′ stem-loop structure, which protects from exonucleolytic degradation. Proteins binding there functions akin to the polyA binding proteins found on other mRNAs to enhance histone mRNA translation by looping back to the 5′ cap structure (*Sanchez and Marzluff, 2002*).

Why *tBip1* mRNA escapes decay remains to be explored. Possible explanations include the presence of secondary structure elements. Indeed, we find a predicted conserved hairpin structure at the 3′ termini of *tBip1* RNAs in some fission yeasts (*S. pombe*, *S. octosporus* and *S. cryophilus*); however preliminary mutational analysis failed to validate its importance for mRNA stability in *S. pombe*. An alternative possibility is that the 3′ end of *tBip1* mRNA may be covalently modified. Such modification would need to be restricted to the 2′-OH, because the 3′-OH group is still accessible for modification by RNA ligase (*Figure 3d*). In this regard, *tBip1* mRNA would resemble miRNAs, which are 2′-O-methylated, conferring resistance to degradation (*Yu et al., 2005*).

Our data show that *Bip1* mRNA is the only mRNA in *S. pombe* in the expressed genome that is subject to this unique regulation. Why would such a singular mechanism have evolved exclusively for Bip1? From work in other species, Bip1 emerges as the most pleiotropically important and precisely controlled ER chaperone (*Gulow et al., 2002*). Moreover, Bip1 holds a unique position in *S. pombe*, where it is glycosylated (*Pidoux and Armstrong, 1993*). A recent comprehensive gene interaction map revealed that Ire1 in *S. pombe* clusters tightly with enzymes involved in the quality control cycle of glycosylated proteins, pointing toward a unique connection between glycosylation and ER stress (*Frost et al., 2012*). By contrast, corresponding E-maps in *S. cerevisiae* succinctly confirm the long-appreciated linear relationship between Ire1 and Hac1 (*Schuldiner et al., 2005*). One may speculate that ER stress in *S. pombe*, as in other species, enhances turnover of glycosylated proteins and that the regulation of *Bip1* mRNA is beneficial to its stability by compensating for such loss (*Bernasconi et al., 2012*).

## Conclusions

From an evolutionary angle, the UPR in *S. pombe* provides an intriguing example of how molecular machines can be repurposed. While input (unfolded proteins, ER stress) and output (RNA cleavage) have been conserved, both detail and global consequences of downstream processes have been adapted to serve different needs. The UPR in both *S. cerevisae* and *S. pombe* fulfills a cytoprotective role, yet the mechanisms of executing this task are opposed. In *S. cerevisiae*, folding capacity is increased via transcriptional up-regulation; in *S. pombe* the folding load is decreased and the ER is restructured. In metazoan cells, both modes of Ire1 activity are merged, and depending on condition and cell type can serve different purposes. RIDD can protect cells by removing major secretory protein loads, as it is the case in insulin secreting cells (*Lipson et al., 2008*), or it can serve to activate apopototic pathways, as it is the case in cells experiencing prolonged and unmitigated ER stress (*Han et al., 2009*).

It has always been puzzling how a strictly cognate system, such as the Ire1- and Hac1-mediated UPR regulatory pathway, would have evolved. While we cannot ascertain what represents the ancestral state, it is tempting to speculate that a broader mRNA degradation pathway preceded the development of the more specialized splicing mechanism. The prevalence of a much broader scope of Ire1 targets in *S. pombe* suggests that a primitive UPR may have served primarily as an ER-localized yet promiscuous RNA degradation system. Individual mRNA substrates would then have evolved appropriate affinities for the enzyme, rendering them more or less susceptible substrates. In this way, the stem/loop splice sites of *HAC1/XBP1* mRNAs could be the result of a long time optimization process: duplication of the cleavage site with concomitant recruitment and repurposing of tRNA ligase would have culminated the UPR splicing reaction. In this view, *S. cerevisiae* emerges as the endpoint of an optimization process rather than an evolutionary precursor. By losing the more ancient RIDD function of the UPR, *S. cerevisiae* cell would have developed to rely exclusively on a more refined and more powerful transcriptional regulation program.

# Materials and methods

## Nomenclature

The unified convention used in this manuscript for genes and proteins is based on the treatment by Alberts, et al. in Molecular Biology of the Cell (5th edition, Garland Publishing), page xxxii. In brief, all genes are denoted in italics with a capitalized first letter. Mutant alleles are indicated by appending a descriptor to the gene name. Proteins are indicated in Roman letters.

## Strains, plasmids and growth conditions

Standard media and genome engineering methods for fission yeast were used as described previously (*Moreno et al., 1991*). Strains and plasmids used in this study are listed in *Table 1*. Briefly, reporter constructs were integrated into the *Leu1* locus using plasmid pJK148. Mutant alleles were integrated by the pop-in/pop-out method using the integrative plasmid pJK210 (*ura4*) (*Guthrie, 2004*). All experiments were carried out in yeast extract complex media (YE5S) or in Edingburgh minimal media (EMM), supplemented with 0.225 mg/ml of L-histidine, L-leucine, L-lysine, adenine and uracil at 30°C, unless otherwise described. For pop-in/pop-out experiments, 5-FOA media containing 1 g/l 5-fluoro-orotic acid was used.

## RNA analysis

Total RNA was purified by standard hot-phenol extraction (*Kohrer and Domdey, 1991*). After precipitation RNA samples were re-suspended in DEPC-treated water and quantified by spectrophotometry. Northern blotting, electrophoresis, labeling, analysis and quantification were performed as described (*Ruegsegger et al., 2001*).

## Protein analysis

Cells were cultured in YE5S media. Between 5 and 10 OD units were collected by centrifugation and snap-frozen. Cell pellets were thawed on ice, re-suspended in 200 µl of lysis buffer (8 M urea, 50 mM HEPES pH 7.4) and lysed in a glass bead mill (5 min at 4°C). After adding 20 µl of a 25% SDS solution, samples were incubated at 65°C for 5 min. The lysates were collected by piercing the bottom of the tubes with a syringe needle and clarified by re-centrifugation (1000 rpm for 10 s). Total protein concentration was determined by a standard bichromic acid (BCA) assay (manual instruction, Pierce Biotechnology). 10 µg of lysate per lane were electrophoresed on SDS-polyacrylamide gradient gels (4–15%, BioRad). The separated proteins were subsequently transferred to PDVF membranes at 200 mA for 1 hr. Blots were blocked with 5% milk in Tris-buffered saline (10 mM Tris, 150 mM NaCl, 0.1% Tween-20) and incubated with primary antibodies overnight at 4°C. Antibodies and dilutions: rabbit polyclonal anti-Kar2 (1:5000), mouse monoclonal anti-RNA polymerase II carboxy-terminal domain (CTD) repeat (Abcam ab817) (1:8000 dilution). Blots were washed and incubated with Li-Cor fluorescently-coupled secondary antibodies (1:5000) for 30 min. Immunoreactive bands were identified using a Li-Cor infrared scanner, and processed with the Odyssey software package.

## *Bip1* mRNA linker-ligation and 3′ rapid amplification of cDNA ends (RACE)

Total RNA (10 µg) from treated or untreated (2 mM DTT, 1 hr) WT cells were incubated with 25 units of polynucleotide kinase (PNK) to remove 2′,3′-cyclic phosphates for 1 hr at 37°C. After deactivating the enzyme (75°C for 10 min), dephosphorylated RNA was precipitated and re-suspended in 6 µl DEPC-treated water. After denaturing the RNA at 80°C for 5 min, a linker-ligation reaction was performed in the presence of dephosporylated, denatured RNA, T4RNA Ligase II (NEB), RNase inhibitor (40 Units), DMSO, 50% PEG and 1 µg of 5′ pre-adenylated linker (5′AppCTGTAGGCACCATCAAT/3ddC3′, as described (*Ingolia et al., 2011*). Reverse transcription was performed using a reverse complement DNA-oligonucleotide to the linker sequence. The 3′ RACE reaction was performed as described (*Scotto-Lavino et al., 2006*). Nested *Bip1*-PCR products were purified from ethidium-agarose gels and sequenced.

## qPCR analysis of *Gas2* mRNA reporter

Total RNA (1 µg) was reverse-transcribed with random hexamers. 1% of the resulting cDNAs was employed for real-time PCR reactions utilizing SYBR green. The reactions were run and analyzed in a DNA Engine OPTICON 2 using the BioRad Opticon Monitor 3.0 software.

**Table 1.** Strains and plasmids used in this study

| Strain | Description | |
|--------|-------------|---|
| yPK001 | WT | |
| yPK002 | Ire1Δ | |
| yPK003 | Ire1(H1018N) | |
| yPK004 | Ski2Δ | |
| yPK005 | Ski2Δ Ire1Δ | |
| yPK006 | Bip1 (CTCGTG\C) | |
| yPK007 | Bip1 (CTGGTC\C) | |
| yPK008 | Bip1 (CTGGTT\C) | |
| yPK009 | Bip1 (ΔACTGGTG\C) | |
| yPK010 | Bip1 (ΔTTAACTGGTG\C) | |
| **Plasmid** | **Description** | **Marker** |
| pPK001 | pJK148::pNda2::Gas2::Nda2 3′UTR | *Leu1* |
| pPK002 | pJK148::pBip1::GFP::Nmt1 3′UTR | *Leu1* |
| pPK003 | pJK148::pNmt1+5′UTR::Bip1(ORF)::Bip1 3′UTR | *Leu1* |
| pPK004 | pJK148::pNmt1::ss(Bip1)::GFP::Bip1 3′UTR | *Leu1* |

## Growth assay and colony forming unit assay for *Bip1* 3′ UTR mutant

WT and *Bip1(ΔTTAACTGGTGC)* mutant cells were grown overnight in (YE5S) media. The next morning cultures were diluted and grown until they reached an OD of 0.25. ER-stress was induced with tunicamycin as indicated. After 3 hr, stressed cells were washed four times with pre-warmed YES5 media to remove the drug, the culture was readjusted to an OD of 0.25, and incubated in a 24 well plate (1 ml per well) over 32 hr at 32°C. The OD was measured every 10 min in a microplate reader (Synergy 4 BioTeK). The colony forming unit assay was performed by plating washed cells at different dilutions on solid media (YE5S) and incubated for 3 days at 30°C. Colonies were counted from the dilutions series. Untreated cells served as a control.

## *Bip1* mRNA half-life measurement

To determine the half-life of *Bip1* mRNA, we constructed an integrative plasmid (pJK148) containing the open reading frame and 3′ UTR of *Bip1* mRNA under the control of a thiamine-regulated *Nmt1* promoter. The construct included the *Nmt1* 5′ UTR. Cells were grown in EMM complete media (without any thiamine) for 24 hr. Cells were then diluted into fresh EMM complete media and re-grown to an OD of 0.3. After inducing ER stress with tunicamycin (0.25 µg/ml) for 1 hr, thiamine was added to 15 µM to block transcription of the *Nmt1* promoter (**Maundrell, 1990**). Samples were processed at indicated time points and subjected to Northern blot analysis. A DNA probe complementary to the 5′ UTR of nmt1 was used for detection.

## RNA-deep-sequencing sets

Figure 1—source data 1:

- oligo-dT-enriched mRNA set I (including: WT, WT +DTT (2 mM, 1 hr) and *Ire1Δ*+DTT (2 mM, 1 hr))
- oligo-dT-enriched mRNA set II (including: WT, WT +DTT (2 mM, 1 hr) and (2 mM, 1 hr))
- total (-depleted rRNA) RNA set I (including: WT, WT +DTT (2 mM, 1 hr)
- total (-depleted rRNA) RNA set II (including: WT, WT +DTT (2 mM, 1 hr) and *Ire1Δ*+DTT (2 mM, 1 hr))

Figure 2—source data 1:

- 2′,3′-cyclic phosphate 3′end mapping setI (*Ski2Δ* (2 mM DTT, 30 min) *Ire1Δ Ski2Δ* (2 mM DTT, 30 min))

## RNA isolation for deep-sequencing library

As indicated, one of three methods was utilized to isolate RNA with specific chemical properties: polyA$^+$ tail enrichment, rRNA depletion, or 3' end 2',3'-cyclic phosphate enrichment. Total RNA was prepared from cells by the hot acid phenol method (*Kohrer and Domdey, 1991*), and subsequent enrichment performed. PolyA$^+$ mRNA was purified by two sequential rounds of enrichment using oligo-dT DynaBeads (Invitrogen) according to the manufacturer's instructions. rRNA depletion was performed by first depleting all abundant RNAs smaller than 200 nt using the modified protocol for isolating only large RNAs provided in the mirVana miRNA Purification Kit (Ambion) followed by two rounds of subtractive hybridization using the Ribominus Eukaryote Kit for RNA-Seq (Invitrogen), according to the manufacturer's instructions. To sequence 2',3'-cyclic-phosphate cleavage products purified tRNA ligase (a kind gift from J.R. Hesselberth) was used to selectively ligate an RNA linker to all 2',3'-cyclic phosphates in total RNA as previously described (*Schutz et al., 2010*). PolyA$^+$-enriched and rRNA-depleted samples were randomly fragmented under basic conditions, precipitated by standard methods, and ~50 nt fragments were size-selected by polyacrylamide gel electrophoresis as described in *Ingolia et al. (2009)*.

## Ribosome footprint isolation for polysome profile and deep-sequencing library

Ribosome footprints were isolated as described in *Ingolia et al. (2009)*, with minor modifications. Briefly, 750 ml mid-log yeast cultures (± 30 min 2 mM DTT treatment) were harvested by filtration without the addition of cycloheximide, and were immediately flash frozen. Frozen cells were cryogenically lysed in the presence of 3 ml of frozen polysome lysis buffer (20 mM Tris pH 8.0, 140 mM KCl, 1.5 mM MgCl$_2$, 100 µg/ml cycloheximide, 1% Triton) on a Retsch MM301 mixer mill, and thawed lysates were subsequently clarified by centrifugation as described. ~50 A$_{260}$ units of clarified lysate was treated with 750 U of *E. coli* RNase I (Ambion) for 1 hr on ice to minimize 80S degradation. Monosomes were collected from sucrose density gradients (10–50% wt/vol, prepared in polysome lysis buffer: 20 mM Tris pH 8.0, 140 mM KCl, 5 mM MgCl$_2$, 100 µg/ml, cycloheximide, 0.5 mM DTT, 20 U/ml SUPERase·In) as described, and undigested control samples were loaded to generate polysome profiles shown in *Figure 4b*. RNA from monosome or polysome fractions was isolated using the hot acid phenol method. For ribosome profiling, 28–34 nt RNA fragments from the monosome fraction were size-selected by gel electrophoresis as described above.

## Library preparation

Deep sequencing libraries were constructed as described in *Ingolia et al. (2011)*. Briefly, size-selected mRNA (polyA$^+$ and -rRNA) or ribosome footprints were 3'-dephosphorylated with T4 polynucleotide kinase (NEB). 3'-dephosphorylated RNA was ligated to a preadenylated miRNA cloning linker (IDT, Linker 1) using T4 RNl2(tr) (NEB) rather than enzymatically polyadenylated. Subtractive hybridization of rRNA contaminants was not performed. Ligated samples were directly reverse transcribed using SuperScriptIII (Invitrogen), circularized using CircLigase (Epicenter), and PCR amplified.

## General RNA-sequencing data processing

Data from deep-sequencing analyses of total (-ribosomal RNA) RNA, polyA$^+$ enriched mRNA and ribosome foot-printing were collected and the resulting sequences were aligned using the following method: the linker sequences at the 3' ends were removed prior to alignment using SOAP2.20, allowing a maximum of two total mismatches (*Li et al., 2009*). Ribosome footprint reads were assigned to a specific A-site nucleotide by an offset of +15 from 5' end of the read (only for ribosome foot-printing data set). All reads aligned to rRNA and tRNA were removed. To align intron-exon junction reads, all reads with no alignment against *S. pombe* genomics sequences were re-aligned against a sequence library of *S. pombe* processed protein-coding transcripts. All the alignments were performed against the most recent version of the *S. pombe* genome (the www.genedb.org/genedb/pombe/). The raw sequencing data will be available for download at NCBI GEO.

## Quantification of mRNA abundance

Biological replicates (set I and set II, where available) were combined to increase read coverage. After combining sets, mRNA (ORF) transcripts with fewer than 100 reads before normalization were excluded. Next, the passed mRNAs (ORF) were normalized to reads per million total reads (rpM). Because the total number of reads may not reflect the total RNA production correctly (*Robinson and Oshlack,*

**Table 2.** Ribosomes footprints of mRNAs encoding protein predicted to enter the ER

| All ER-targeted mRNAs ribosome occupancies (N=1014 mRNAs) | rpkm | Ration normalized to WT-DTT | % |
|---|---|---|---|
| WT (-DTT) | 83,540 | 1 | 100 |
| WT(+DTT) | 71,996 | 0.8618147 | 86.18146995 |
| *Ire1Δ*( +DTT) | 90,166 | 1.079315298 | 107.9315298 |

rpkm: reads per kilo-base of transcript per million reads.

*2010*), the observed count for gene *g* in condition *k* need to be normalized to the total RNA production to calculate the fold-change between two conditions. We used the following method to estimate the correct RNA abundance for given RPM values. Given conditions *k* and r, we calculated the expected RNA abundance of condition *k* given the RPM value in condition r with the following equation: $x_{k,r,g} = 2^{(a_{k,r} \log_2(x_{r,g}) + b_{k,r})}$.

$x_{r,g}$ is the RPM value for gene *g* in the reference condition, $x_{k,r,g}$ is expected RPM estimated with linear regression between the RPM values from condition *k* and *r*. $a_{k,r}$ and $b_{k,r}$ are the coefficients of the linear regression where $a_{k,r} = \dfrac{\sum(Y_{k,g} - \bar{Y}_k)(Y_{r,g} - \bar{Y}_r)}{\sum(Y_{r,g} - \bar{Y}_r)(Y_{r,g} - \bar{Y}_r)}$, $b_{k,r} = \bar{Y}_k - a_{k,r}\bar{Y}_r$ in which $Y_{k,g} = \log_2(x_{k,g})$, $Y_{r,g} = \log_2(x_{r,g})$.

For the above linear regression, genes with $M_g = \dfrac{x_{k,g}}{x_{r,g}} \geq 10$ or $\geq 0.1$ were removed from the data set to estimated $a_{k,r}$ and $b_{k,r}$. The fold change between condition *k* and *r* is then calculated as: $F_{g,k,r} = \dfrac{x_{k,g}}{x_{k,r,g}}$, which is the RPM value for gene *g* under condition *k* divided by the expected RPM estimated above.

## 2′,3′-cyclic phosphate 3′ end RNA mapping and Ire1 cleavage site motif determination

Sequence read alignments from the 2′,3′-cyclic phosphate mapping data were performed as for the RNA-Seq reads described above. The design of the library (3′ end mapping) made it necessary to align the reads to the opposed strand of the gene. To map putative Ire1-dependent cleavage sites with high stringency, we identified the positions within each transcript containing more than four reads in RNA derived from the *Ski2Δ* sample and zero reads in RNA derived from the *Ire1Δ Ski2Δ* sample. We identified by this method 4027 putative Ire1-dependent cleavage sites in the genome. By using a less stringent criterion (by taken the ration between *Ski2Δ* sample and *Ire1Δ Ski2Δ* sample: we allowed a ratio of 5 or more). By this criterion we identified 4134 putative Ire1-dependent cleavage sites. The overlap between the two methods is 97%. We continued our analysis with the more stringent criteria of only allowing zero reads in *Ire1Δ Ski2Δ* sample. To identify an overrepresented consensus sequence, we first extended the sequences 9 nt upstream and downstream of the potential cleavage sites. Furthermore, we used a position weight matrix (PWM), which was generated from these sequences by weighting each sequence with the reads from the *Ski2Δ* sample. By using all annotated genes and the corresponding putative cleavage sites, we could not identify overrepresented sequences. The same held true for the ER-target mRNAs set (N=1014). By contrast, a strong consensus motif (*Figure 2e*) emerged when we mapped the putative cleavage sites in our set of 39 Ire1- and ER stress-dependently twofold down-regulated transcripts.

## Estimation of protein flux into the ER

We scored ribosome footprint reads per kilobase (to normalize for length of open reading frames) per million reads (to compare different conditions) for the set of mRNAs, encoding proteins predicted to enter the ER as described above (see *Table 2*).

## Acknowledgements

We thank Diego Acosta-Alvear, Brooke Gardner, Benoît Kornmann, David Pincus and colleagues for their help and discussion. We also thank the members of the Walter lab for critical comments on the manuscript. We are grateful to S. Forsburg and N. Krogan for providing strains and plasmids, and to J.R. Hesselberth for the kind gift of purified tRNA ligase, reagents and valuable advice. We thank the UCSF CAT facility in particular Clement Chu for assisting with sequencing.

## Additional information

### Funding

| Funder | Author |
|---|---|
| Howard Hughes Medical Institute | Peter Walter |
| National Science Foundation | Marcy Diaz |

The funders had no role in study design, data collection and interpretation, or the decision to submit the work for publication.

### Author contributions

PK, Conception and design, Analysis and interpretation of data, Drafting or revising the article; MD, Conception and design, Analysis and interpretation of data, Drafting or revising the article; JZ, Analyzed the genomics data; performed statistical analysis; CCW, Conception and design; AL, Conception and design; TA, Performed bioinformatics analyses; HL, Analyzed the genomics data; performed statistical analysis; PW, Conception and design, Analysis and interpretation of data, Drafting or revising the article.

## Additional files

### Major datasets

The following datasets were generated

| Author(s) | Year | Dataset title | Dataset ID and/or URL | Database, license, and accessibility information |
|---|---|---|---|---|
| Kimmig P, Diaz M, Zheng J, Williams C, Lang A, Aragón T, Li H, Walter P | 2012 | The unfolded protein response in fission yeast modulates stability of select mRNAs to maintain protein homeostasis | GSE40298; http://www.ncbi.nlm.nih.gov/geo/query/acc.cgi?acc=GSE40298 | In the public domain at GEO: http://www.ncbi.nlm.nih.gov/geo/ |

**Reporting standards:** Microarray data were deposited in the NCBI Gene Expression Omnibus (GEO) repository

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
