## [Decision Letter]

Thank you for choosing to send your work entitled ‘The Unfolded Protein Response in Fission Yeast Modulates Stability of Select mRNAs to Maintain Protein Homeostasis’ for consideration at *eLife*. Your article has been evaluated Roy Parker (Reviewing Editor) and Randy Schekman (Editor-in-Chief). We are pleased to provisionally accept this most interesting work for publication in *eLife*.

We have assembled the following comments. Our goal is to provide the essential revision requirements as a single set of instructions, so that you have a clear view of the revisions that are necessary for us to publish your work. In this case, the comments are minor but addressing them would further strengthen the submission.

**General assessment:**

This manuscript addresses the mechanism of the ER UPR in *S. pombe*. The major contribution of the work is to show that the response to UPR in *S. pombe* is dominated by an IRE dependent degradation of mRNAs, as well as a novel 3′ UTR processing of the BIP mRNA, which stabilizes the mRNA thereby leading to more BIP protein being produced.

We recommend publication of the work for two reasons.

A) This work is significant in two manners: it reveals a novel and unexpected diversity of how cells can manage the UPR, and it also reveals a novel form of RNA processing wherein the cleaved and truncated mRNA is more stable, which is unexpected and will be sure to yield future mechanistic insights as that processing is studied further.

B) The work is well done, the manuscript clearly written, and the conclusions well supported.

**Specific comments for consideration in a revised submission:**

1) The conclusion that the mRNAs are endonucleolytically cleaved is based on the detection of the expected 5′ mRNA fragment in a Ski2∆ strain (which blocks 3′ to 5′ degradation). Typically, endonuclease sites are demonstrated by also showing the 3′ mRNA fragment accumulates in an Xrn1∆ strain. Such a demonstration would not be required for publication but would strengthen the work.

2) It might be worth pointing out that BIP mRNA does not appear to be cleaved by RIDD in mammals so this aspect of UPR is unique (to date) to Pombe.

3) Another explanation for why 15% reduction in protein flux into the ER would have a disproportionate impact is that the mRNA cleavage occurs in cis on mRNAs where the co-translationally expressed protein is misfolding.

---

## [Author Response]

Specific response to Reviewers' comments:

*1) The conclusion that the mRNAs are endonucleolytically cleaved is based on the detection of the expected 5′ mRNA fragment in a Ski2∆ strain (which blocks 3′ to 5′ degradation). Typically, endonuclease sites are demonstrated by also showing the 3′ mRNA fragment accumulates in an Xrn1∆ strain. Such a demonstration would not be required for publication but would strengthen the work*.

An excellent suggestion and we had considered the experiment in the course of this work. Xrn1-deleted S. pombe cells grow very poorly, however, even in the absence of ER stress. Moreover, we feel that the issue of endonucleolytic cleavage will be much cleaner addressed in in vitro experiment, where a clear precursor/product relationship for the substrate can be ascertained. Such experiments are in progress in the lab, and will be part of a more extensive mechanistic dissection of the reaction.

*2) It might be worth pointing out that BIP mRNA does not appear to be cleaved by RIDD in mammals so this aspect of UPR is unique (to date) to Pombe*.

We agree and have added a statement to clarify this point.

*3) Another explanation for why 15% reduction in protein flux into the ER would have a disproportionate impact is that the mRNA cleavage occurs in cis on mRNAs where the co-translationally expressed protein is misfolding*.

We agree and have edited the Discussion accordingly.